



# A rapid quantitative screening method to assess chemicals present in heated e-liquids and e-cigarette aerosols

Natalie Anderson [1,2], Paul Pringle [3], Ryan Mead-Hunter [1], Benjamin Mullins [1], Alexander Larcombe [1,2], Sebastien Allard [3]

[1]School of Population Health, Curtin University, Perth, Australia, [2]Wal-yan Respiratory Research Centre, Telethon Kids Institute, Nedlands, 6009, Australia, [3]Curtin Water Quality Research Centre, School of Molecular and Life Sciences, Curtin University, Perth, Australia

*Correspondence to*: Natalie Anderson (Natalie.anderson@telethonkids.org.au)

**Abstract.** Introduction: Electronic cigarettes (e-cigarettes) lack regulatory status as therapeutic products in all jurisdictions worldwide. They are potentially unsafe consumer products, with significant evidence showing they pose a risk to human health. Therefore, developing rapid, economical test methods to assess the chemical composition of e-liquids in heated and unheated

forms and the aerosols produced by e-cigarettes is crucial. Methods: Four different e-liquids were heated using two different methods: 1) "typical" vaping using an e-cigarette device, by cycling "on" for three seconds every minute for two hours (e-liquid obtained from remainder in the tank and aerosol collected in an impinger) and, 2) "accelerated" heating, using an e-cigarette coil, submerged in e-liquid, and heating in short 20 second bursts "on" then 20 seconds "off" for two minutes only (liquid traps aerosol produced). All e-liquids were then analyzed to test for the presence and quantity of 13 chemicals by gas-

chromatography mass-spectrometry and compared to an unheated sample. Results: E-liquids heated with the "accelerated" method showed a comparable trend to the "typical" heating method, of increase or decrease in chemical compound quantity, for greater than two-thirds of the detected compounds analyzed over all e-liquids. Six chemicals were detected as aerosol from the impinger fluid with the "typical" heating method, most at negligible levels. Conclusion: We propose this rapid method could form the basis of a standardized screening tool to test heated e-liquids (and e-cigarette aerosols) for harmful or banned

substances to ensure only approved products reach the consumer, and the potential harms of e-cigarette use are reduced.

## 1 Introduction

The potential for e-cigarettes to negatively impact health is of concern due to the known presence of unsafe chemical constituents, and the possibly carcinogenic, mutagenic or reprotoxic nature of the aerosols that are produced by heating, aerosolizing and inhaling e-liquids (Goniewicz et al., 2014, European Parliament, 2014). International Agency for Research

on Cancer group one carcinogens, such as acetaldehyde and formaldehyde have been found in e-cigarette aerosols and are known degradation species of the main e-liquid components, propylene glycol and glycerin (Goniewicz et al., 2014). Many other ingredients that pose a risk to human health, or for which the inhalation health effects are unknown, are regularly found



in unheated e-liquids, occasionally at unsafe levels – most are ingredients added to the e-liquids as flavors or solvents (European Association for the Co-ordination of Consumer Representation in Standardisation, 2021). These potentially unsafe

ingredients may be present in unheated e-liquid, heated e-liquid, aerosol generated by the heating process, or in any combination of the three forms. However, as recently as 2015, no country in the world regulated e-liquid ingredients beyond nicotine levels. As more evidence emerges suggesting that e-cigarettes negatively impact health, regulation is rapidly evolving in this area (European Parliament, 2014, Budzyńska, Sielemann, Puton, & Surminski, 2020, Therapeutic Goods Administration, 2021). For example, the European Union (EU) Tobacco Product Directive (TPD) states that only ingredients

in nicotine-containing e-liquids that do not pose a risk to human health in heated or unheated form can be used (European Parliament, 2014). As a result of the EU TPD, countries that have banned ingredients include the United Kingdom, Germany and France (European Parliament, 2014, European Association for the Co-ordination of Consumer Representation in Standardisation, 2021, Budzyńska, Sielemann, Puton, & Surminski, 2020). A modest ban on certain ingredients, in nicotine-containing e-liquids only, also came into effect in October 2021 in Australia, and a complete ban on non-prescription e-

cigarette importation and sale in Australia was announced in May 2023 (Nogrady, 2023, Therapeutic Goods Admnistration, 2021). The concerns for e-liquids (heated and unheated) and their aerosols, to negatively impact health are amplified by unregulated e-liquid use, a multibillion-dollar market driving product sales and growth, emerging evidence of self-reported lung conditions associated with e-cigarette use, and E-cigarette or Vaping Associated Pulmonary (or Lung) Injury (aka VAPI, or VALI/EVALI) (Greenhalgh, 2019, Osei et al., 2020, Bircan, Bezirhan, Porter, Fagan, & Orloff, 2021).

Whilst chemical testing of unheated e-liquids is relatively common, the safety of e-liquids remains largely unassessed due to the sheer scale of the market, and with the exception of the EU TPD guidance, to the best of our knowledge, testing heated e-liquids is not required in Australia or elsewhere (European Parliament, 2014, Greenhalgh, 2019, Scientific Committee on Health Environmental and Emerging Risks 2021, Larcombe et al., 2021). Methods generally involve testing of unheated e-liquids or the e-cigarette aerosol generated, but not the heated e-liquid (Scientific Committee on Health Environmental and

Emerging Risks, 2021). This is important however, as heated e-liquids are more representative of what the user inhales, compared to unheated e-liquids, but also easier to assess when compared to e-cigarette aerosols (Larcombe et al., 2021, Erythropel et al., 2019). Further, chemicals present in e-liquids are known to degrade due to heating and form secondary products that may have increased (or decreased) toxicity compared to the parent compound (Goniewicz et al., 2014, Erythropel et al., 2019). For example, the degradation processes for e-liquid base components propylene glycol and glycerin, have been

reported for both low (< 200ºC) and high temperatures (>200ºC) and the physical mechanisms underlying aerosol production (and therefore chemical formation) have been described to be either by boiling or evaporative-convection depending on wetted-wick temperature (Jaegers, Hu, Weber, & Hu, 2021, Floyd, Queimado, Wang, Regens, & Johnson, 2019, Li et al., 2021, Zhao, Shu, Guo, & Zhu, 2016, Saliba et al., 2018). For a given e-liquid, the abundance of degradation products depends on numerous factors including the temperature the coil is heated to, the availability of oxygen, and the exposure to a potentially catalytic

surface such as Kanthal (iron-chromium-aluminum alloy) (Saliba et al., 2018). Kanthal is commonly used in e-cigarette heating coils, and it has been shown to reduce the temperature needed to thermally decompose e-liquid components (Jaegers, Hu,



Weber, & Hu, 2021). Many approaches have been trialled to test the chemical composition of e-liquids since the introduction of the modern e-cigarette in 2003. Current methods have been summarized recently and shown to be outdated or limited (Scientific Committee on Health Environmental and Emerging Risks, 2021). Therefore, it is critical to standardize the

procedures allowing to test the chemical composition of e-liquids. There are many difficulties in directly assessing an e-cigarette produced aerosol. It is challenging to collect enough aerosol to perform an assay and detect potentially toxic compounds, especially because of the overload of the main excipients (propylene glycol and vegetable glycerin). This induces the use of increasingly complicated test methods (Floyd, Queimado, Wang, Regens, & Johnson, 2019, Herrington & Myers, 2015). Despite the expansive range of tests, a simple test capable of assessing heated and unheated e-liquids, and the aerosol

produced all at once is yet to be established, but vital.

In this study we aimed to establish and validate an "accelerated", method of heating e-liquids that would be comparable to, but quicker than, the "typical" vaping of a user and that could assess the heated e-liquid and the aerosol produced all at once. We hypothesized that the "accelerated" method and the commonly used "typical" vaping method would result in similar heating-induced chemical changes in the e-liquids. Comparability of heating method was assessed by measuring the presence

and concentration of the same chemicals with both methods. We hope that this "accelerated" test methodology can form the critical first step in establishing a rapid test for screening of e-liquids for banned substances.

## 2 Materials and Methods

### 2.1 "Typical" vaping process

A set up was designed to replicate the heating/cooling process an e-liquid would undergo when an e-cigarette is used in a

"typical" way (Figure 1) (Etter & Bullen, 2014, St Helen et al., 2016, Cooperation Center for Scientific Research Relative to Tobacco (CORESTA), (2015)). The method allowed sample collection at two points for analysis of heated e-liquid and aerosol respectively: 1) from the remainder in the e-liquid tank (atomiser), and 2) from the impinger (Figure 1). To begin, the e-cigarette (MVP4, Innokin, Shenzhen, China, operating wattage range 6–100 W, temperature range 150–315 ºC, maximum current 35.5A) atomizer was filled with ~3.5 mL of e-liquid and the impinger was filled with 5 mL of e-liquid excipient (50:50

glycerin–propylene glycol (v/v)), Sigma Aldrich, Milwaukee WI, USA). A flow of ~3 L/min ambient filtered air was drawn through the system via laboratory bench vacuum and kept stable through monitoring with a flow meter (Max 5 Lpm, TSI, Shoreview, MN). New coils (Kanthal BVC, 100–200 W, 0.28 Ω, Innokin Scion) were used each time to avoid cross contamination of chemical species and to control for coil ageing effects. The e-cigarette device was set to 80 W (reading 0.28–0.35 Ω) each time the device was connected to the atomiser. To vape the device in line with recommendations by the

Cooperation Centre for Scientific Research Relative to Tobacco (CORESTA), the ignition button was held for ~3 seconds, and the aerosol drawn from the device into a 60 mL syringe and then expelled into the custom made (27 L – 30x30x30 cm) chamber, using a three-way tap (Dispoflex™, Disposafe health and life care Ltd, Haryana, India) (Cooperation Center for Scientific Research Relative to Tobacco (CORESTA) (2015)). This process was repeated every minute for two hours (with



the atomizer tank refilled after ~60 minutes), so that a volume of 7.2 L of e-cigarette aerosol containing air was introduced to the system. After heating, the liquid was transferred to glass sample vials, and kept at 4ºC to minimise the loss of volatiles.

## 2.2 "Accelerated" vaping process

An accelerated ageing/vaping process was developed, based on standard tests for ageing/oxidation of oils (American Society for Testing Materials, 2009). Identical Kanthal BVC coils, as used in the "typical" vaping process, were connected to a power supply (MP3090, PowerTech, China), by means of solid coper wires connected to each end of the coil (end cap removed). The

power supply was set at 7.4 V and 27 A (0.274 Ω) to stay within the maximum power (200 W) of the coil used for the "typical" process and to ensure that the resistance matched that of the "typical" vaping process. The coil was placed in 100 mL beaker, which was open to air, held on a 45º angle with a clamp-stand and ~30 mL of e-liquid was poured into the beaker, enough to completely submerge the coil and ensure the full volume of liquid would not heat to boiling temperature within the one minute total heating period (20 seconds on 20 seconds off x 3), and no planar surface evaporation would occur (Figure 2). The 45º

angle was used both to minimize the liquid volume needed to immerse the coil and to ensure any aerosol produced would recondense on the wall of the beaker allowing it to be collected for sampling. The power supply was then turned on to operate the coil for 5 x 20 second "burst" intervals with 20 second pauses interspersed for a total "on" time of 1 minute, mimicking a "short cluster" vaping pattern for a user (St Helen et al., 2016). After heating, the liquid was transferred to glass sample vials, and kept at 4ºC to minimise the loss of volatiles.

## 2.3 Sample and chemical selection

Four flavoured e-liquids, labelled "nicotine-free", were assessed – "Butterscotch Tobacco", "Menthol", "Choc Caramel", and "Tiramisu" which were purchased from online suppliers and analyzed as 50:50 propylene glycol–glycerin (v/v) ratios. Each e-liquid chemical composition was assessed using both methods to quantify 13 chemicals: 4-(4-methoxyphenyl)-2-butanone, ethyl vanillin, eugenol, nicotyrine, nicotine, menthol, thymol, ethyl maltol, trans-cinnamaldehyde, 2-chloro-phenol, benzyl

alcohol, benzaldehyde, and furfural, with a molecular weight range from 178.23 to 96.09 g/mol. The 13 chemicals were chosen based on (i) being previously identified, known ingredients in e-liquids (ii) the availability of a standard for the chemical (Larcombe et al., 2021).

## 2.4 Chemical analysis method of "Accelerated" and "Typical" vaping process.

Thirteen chemicals were tested for, in four different e-liquids, using gas-chromatography mass-spectrometry. For each of the

four e-liquids, we tested for chemicals in three forms – (i) "unheated" e-liquid (i.e. straight out of the bottle), (ii) remainder of e-liquid in the atomizer and collected from the impinger after "typical" vaping and (iii) e-liquid remaining in the beaker after "accelerated" vaping. The latter sample (iii) was taken in order to detect aerosols, and assuming that aerosols (not vapor) would be captured in the e-liquid with/during the accelerated method. The aerosol generated from the "typical" method was captured in an impinger containing 50:50 (v/v) glycerin–propylene glycol. Our intention was for this collected aerosol in the impinger



to be added to the atomiser tank sample, for equivalent comparison to accelerated sample, however negligible values for the
impinger result meant that they were excluded from the final analysis and are shown in supplementary only (Supplementary
A).

Samples obtained from both methods used to heat e-liquids were compared to unheated e-liquids, both within e-liquid type,
and within chemical compound, with the purpose of the comparison being to identify trends of increase or decrease from
unheated e-liquid.

**2.5 Chemical detection and analysis**

Chemical analysis of "accelerated" and "typical" vaping process e-liquids, including sample and chemical detection, has been
previously described in detail elsewhere (Larcombe et al., 2021). The samples (0.25 g) were accurately weighed and placed
into amber vials with 4.75 mL ultrapure water. Thereafter, 10 µL of a 1 g/L 4-bromophenol-d4 stock solution was added as an
internal standard. Prior to the analysis, 1.6 g of analytical grade sodium chloride was added to increase volatilisation and the
vials tightly capped. To facilitate adsorption, the samples were incubated at 90°C for 15 min prior to solid-phase micro-
extraction using a divinylbenzene/carboxen/polydimethylsiloxane fiber from Supelco® allowing for 13.6 min adsorption of
the analytes on the fiber. The fiber was then desorbed at 250 °C in the injector in spitless mode for 5 min followed by 15 min
in split mode. A Gerstel MPS2 multifunction autosampler was used to perform automated solid-phase micro-extraction
injections. Analysis were carried out with an Agilent 6890N gas-chromatograph interfaced with an Agilent 5973 Network
Mass Selective Detector, fitted with a HP-INNOWax polyethylene glycol stationary phase capillary column (30 m; 0.25 mm;
0.25 µm, Agilent J&W, Australia), to separate polar compounds. A constant flow (1.2 mL.min$^{-1}$) of helium (99.999% pure,
BGC, Australia) was used as a carrier gas. Optimal gas-chromatography mass-spectrometry conditions were determined, as
measured by maximum sensitivity, baseline separation of analytes and gaussian peak shapes. In order to ensure a good
separation of the different analytes, the oven was held isothermal at 37 °C (2 min), then heated to 260 °C at 5 °C.min$^{-1}$, and
held at the final temperature for 10 min. Detection of analytes was carried out using a mass spectrometer in electron impact
ionization mode at 70 eV. The mass spectrometer quadrupole temperature was set at 150°C and the mass spectrometer source
at 230°C. The compounds were identified using a combination of their retention times, comparison of the mass spectra data of
pure compounds and the specific diagnostic ion fragments of each component, with the National Institute of Standards and
Technology Mass Spectral search program from the NIST/EPA/NIH EI and NIST Tandem Spectral Library which came
integrated with the analysis software.

**3 Results**

Over all e-liquids, three of the thirteen compounds tested for were not detected in any e-liquid type (4-(4-methoxyphenyl)-2-
butanone, thymol, 2-chlorophenol) (Table 1).





### 3.1 Inclusion and exclusion criteria for analysis


There were 16 instances where a chemical was detected in unheated and both heated forms (Table 1) and all were included in analysis. Analysis involved: 1) simple comparison in table format of the heated (two methods) and unheated form of an e-liquid sample and, 2) comparison via fold change compared to unheated ((Y-X)/X, where X is the unheated sample (mg/L concentration) and Y is the heated sample (mg/L concentration)) for both "typical" and "accelerated" heating methods.


Analyses were not possible on the following: one chemical was undetected in unheated form but detected in both heated forms (menthol in butterscotch tobacco); one chemical was undetected in unheated form and detected in only one heated form (trans-cinnamaldehyde in choc caramel). A further three chemicals were detected in unheated form and only one heated form (benzyl alcohol (Tiramisu), eugenol (Tiramisu) and furfural (Tiramisu)) (Table 1).

### 3.2 Behaviour of the different chemicals detected in e-liquids (Figure 3)


To compare the effect of heating, results are displayed as fold change compared to unheated ((Y-X)/X, where X is the unheated sample (mg/L concentration) and Y is the heated sample (mg/L concentration)) for both "typical" and "accelerated" heating methods (Figure 3). Specific chemicals (benzaldehyde, benzyl alcohol, ethyl vanillin, ethyl maltol, furfural, menthol, nicotine, and nicotyrine) were present in unheated form, and both heated forms in 16 instances.

Over all e-liquid types in these 16 instances, 70% (11/16) demonstrated a consistent trend within chemical type i.e. both


methods of heating either increased or decreased in concentration compared to unheated sample. Ethyl vanillin (choc caramel and tiramisu), furfural (butterscotch tobacco), ethyl maltol (tiramisu), and benzaldehyde (choc caramel) are the five exceptions.

### 3.3 Chemical characterization by e-liquid type

In the "Menthol" e-liquid, of the 13 chemicals tested, nine were not detected in the heated or unheated sample (Table 1). Of the four that were detected (nicotine, nicotyrine, menthol and benzaldehyde), all (4/4, 100%) exhibited the same trend


(increasing concentration) after heating when compared to the unheated sample for both "typical" and "accelerated" heating methods (Table 1, Figure 3).

In the "Butterscotch Tobacco" e-liquid, of 13 chemicals tested, eight were not detected in heated or unheated form, and an additional one (menthol) was undetected in unheated form (Table 1). Of the four chemicals detected in each sample, three (3/4, 75%) (ethyl vanillin, benzyl alcohol and benzaldehyde) exhibited the same trend for both "typical" and "accelerated" heating


methods when compared with the unheated e-liquid (Table 1, Figure 3). However, furfural increased after "typical" heating but decreased with the "accelerated" method when compared to the unheated sample.

In the "Tiramisu" e-liquid, of 13 chemicals tested, seven were not detected in heated or unheated form, and an additional three were undetected in one form of heating (Table 1). Of the three detected in each sample, one (1/3, 33%) (benzaldehyde) exhibited the same trend after both "typical" and "accelerated" heating (Table 1, Figure 3). The remaining two chemicals



detected (ethyl vanillin and ethyl maltol) both decreased after "typical" heating but increased after "accelerated" heating when compared to the unheated sample.

In the "Choc Caramel" e-liquid, of 13 chemicals tested, seven were not detected in heated or unheated form and an additional one (trans-cinnamaldehyde) was undetected in both unheated and heated form (Table 1). Of the five chemicals detected in each sample, three (3/5, 60%) (benzyl alcohol, ethyl maltol, and furfural) exhibited the same trend after both "typical" and

"accelerated" heating methods (Table 1, Figure 3). Benzaldehyde increased after "typical" heating but decreased after "accelerated" heating compared to the unheated sample. Conversely, ethyl vanillin increased after "accelerated" heating, but decreased after "typical" heating when compared to the unheated sample.

### 3.4 Impinger results from "typical vaping" heating method

Only six of the 13 chemicals tested were detected in the impinger fluid; furfural, benzaldehyde, menthol, benzyl alcohol, ethyl

maltol and ethyl vanillin. However, as the impinger results were negligible, compared to heated e-liquid, for most compounds, results from the impinger fluid have been included only in supplementary material (Supplement A).

### 4 Discussion

The "accelerated" method used here is simple, cost effective and has the potential to produce heated e-liquid and aerosol for chemical assessment in a single experiment. Many other simplified "in house" methods and set-ups (e.g. e-cig puffing

machines) exist for vaping, due to the prohibitive costs of commercially available vaping machines, but to the best of our knowledge, they all focus on generation of the e-cigarette aerosol and not on assessment of the heated e-liquid (Palazzolo, Caudill, Baron, & Cooper, 2021).

Comparison of the accelerated and typical heated samples with their unheated counterpart showed, overall, that over two-thirds (~70%) of the results (in 11 out of the 16 total chemical comparisons), the methods demonstrated a similar trend (increase or

decrease) while in one-third (~30%) of the results (5 out of 11 comparisons) a different trend was observed between heating methods. The four chemicals implicated when different trends were observed were mostly aldehydes (ethyl vanillin (2/11), furfural (1/11), and benzaldehyde (1/11)) except for ethyl maltol (1/11) being an alcohol. Three of these five differences (ethyl vanillin (2/11) and ethyl maltol 1/11) demonstrated an increase in chemical concentration with the "accelerated" heating method compared to the "typical" method. The observed "increase" with the "accelerated" method can be attributed to a

decrease in chemical quantity with the "typical" method, due to loss of aerosol with our "typical" vaping experimental method, as evidenced by rainout (recondensation of the aerosol as it cools) of the liquid aerosol in the three-way tap system and tubing (ID < 5 mm) connecting the tap system to the 27 L chamber (Figure 1). Rainout is likely to contribute to the reduction, considering that only six of the 13 chemicals tested for were detected at all in the impinger fluid (typical vaping set-up), and these six included the four chemicals where the trend (to increase or decrease from baseline) differed between heating methods

(ethyl vanillin, ethyl maltol, benzaldehyde, and furfural), albeit at very low levels. We suspect that with a modified design,



rainout losses could be reduced or sampled, thus increasing the yield from the impinger with the "typical" method and also improving comparability between methods. It is also possible that the flavor aldehydes were present in their propylene glycol acetal form instead of their aldehyde form, as aldehydes are known to form acetals readily (Erythropel et al., 2019). The inability to fully capture the aerosol from the "typical" heating method due to rainout, meant we are unable to confirm the

suitability of our "accelerated" heating method as an impinger for aerosol, but only its validity to compare heating methods. However, it is likely to be suitable considering that the data were consistent ~70% of the time, and differences can be mostly (in 3/5 instances) explained by losses to rainout.

The remaining two (out of five) discrepancies involved furfural (1/11) and benzaldehyde (1/11), and these compounds were found in increased quantities with the "typical" method compared to the "accelerated" method. Whilst it is suggested that

solubility of the flavoring compound in the parent compound, and not the boiling point, is the major indicator of whether or not a compound will be detected in/carried over into an aerosol, lower quantities when using the "accelerated" method may still be partially explained by the low molecular weight and low boiling point of these compounds (Erythropel et al., 2019). For example, furfural has the lowest boiling point and molecular weight of all chemicals detected (162℃, 96.09 g/mol) and benzaldehyde the second lowest (178.1℃, 106.12 g/mol), meaning it is plausible they may volatize/form vapor (not aerosol)

more readily and be lost as vapor if the wetted wick temperature was different between methods (perhaps increased with accelerated method). Our study design angled the beaker at 45° to allow re-condensation of any vapor on the beaker wall, however the experiment was carried out in a fume hood for health and safety, which may have assisted vapor removal. In future, putting a lid on the angled beaker, monitoring the wetted wick temperature of the coil in both methods (not just coil temperature), the liquid temperature, or other parameters such as in previous studies, would elucidate further mechanisms

behind these discrepancies, and allow full validation of the method for detection of aerosols (Li et al., 2021, Palazzolo, Caudill, Baron, & Cooper, 2021, Bitzer et al., 2018).

Undetected chemicals included 4-(4-methoxyphenyl)-2-butanone, thymol and 2-chloro-phenol. Considering no "fruity" flavors were assessed, it is less surprising that 4-(4-methoxyphenyl)-2-butanone was undetected, as it is a raspberry ketone methyl ether – a common flavoring in "berry" flavored e-liquids. Thymol (a phenol and monoterpenoid) is a flavor of its own,

but synthetic thymol (produced from m-cresol) is also used as a precursor to produce racemic menthol, as is pulegone and other terpenoids, and so perhaps thymol was not the precursor for menthol in those e-liquids and therefore not a contaminant (Dylong, Hausoul, Palkovits, & Eisenacher, 2022). For example, as the demand for menthol increases, alternative methods to produce L-menthol are on the rise, such as from citronellal (Dylong, Hausoul, Palkovits, & Eisenacher, 2022). Additionally, there are synthetic analogues to menthol such as N-ethyl 2-isopropyl-5-methylcyclohexanecarboxamide (trade name WS-3).

Menthol was not detected in either tiramisu or choc-caramel flavors so perhaps either synthetic analogues were present (where menthol was not detected), or the production method for the menthol was not using thymol as an intermediary. Chlorophenols like 2-chloro-phenol have previously been detected in e-liquids probably because they are notorious environmental contaminants – 2-chloro-phenol is a priority contaminant in both the US and EU and has previously been found in e-liquids (Larcombe et al., 2021, Chivers, Janka, Franklin, Mullins, & Larcombe, 2019, Igbinosa et al., 2013). Two-chloro-phenol is



used for many applications, predominantly its role as a detergent, however, other roles for 2-chlorophenol includes its use as
an intermediate in the manufacturing of agricultural chemicals, pharmaceuticals, biocides, and dyes, thus it is commonly
detected in environmental water samples after being discharged from industrial effluents (Igbinosa et al., 2013, Yahaya, Okoh,
Agunbiade, & Okoh, 2019). It has been previously suggested that 2-chloro-phenol may be a contaminant from the glycerin
excipient, for two reasons; 1) vegetable glycerin is made from plant crops such as canola and 2-chlorophenol has been found
in canola as a pesticide residue and, 2) glycerin (not from plants) is a by-product of bio-diesel production and biodiesel can be
made with canola (Abdel-Gawad, H., & Hegazi, B., 2010, Yahaya, Okoh, Agunbiade, & Okoh, 2019). It is possible that lesser-
known/detected derivatives of 2-chloro-phenol or other phenolic derivatives known to be priority contaminants were present.
However, it was not within the scope of the study to assess these as they are not commonly reported to be found/tested for in
e-liquids. Eugenol and trans-cinnamaldehyde were the least detected compounds, they were found in only one flavor e-liquid.
Trans-cinnamaldehyde was only detected in a single accelerated sample, which is perhaps expected as cinnamaldehyde is less
commonly found, however it could also be absent as the acetal form of the parent aldehyde is known to be more reactive (αβ,
unsaturated aldehyde) and to degrade to secondary compounds more readily (Erythropel et al., 2019).

**4.1 Limitations and future directions**

A limitation of this study is that we tested a pre-determined list of chemicals, based on our knowledge of known e-liquid
ingredients, available standards and available analytical methods, rather than obtaining a complete chemical characterization.
This approach allowed us to test a larger range of e-liquids and demonstrates the utility of the "accelerated" aging technique,
as per the over-arching goal of the study. However, an "open ended" approach may be useful for future studies if this method
is to become standardized. An open-ended approach would allow a more complete comprehension of the aging process and
oxidation reactions occurring but would require a broader range of analytical techniques than demonstrated here. Furthermore,
whilst we assessed for some ingredients which are now banned (i.e. benzaldehyde and cinnamaldehyde), this study was
designed and conducted prior to the enactment of banned ingredients by the Therapeutic Goods Administration in 2021
(Therapeutic Goods Administration, 2021) and expanding the analysis to include these chemical products would be helpful in
future studies. The same comment applies for chemicals banned in other jurisdictions.

The method we describe in this study has many advantages over current methods for testing e-liquids. It is a rapid and
inexpensive set-up allowing assessment of the chemical composition of heated e-liquids and, potentially, their resultant
aerosols. It could be used with any available coil that can be modified and powered as described. Furthermore, the accelerated
method is likely to capture aerosol generated from a heated e-liquid in a manner comparable to the "typical user" vaping
method as described in CORESTA in terms of both type and quantity of chemicals produced. Our submerged, rapid heating
and cooling method is able to economically sample heated liquid and aerosols (but not vapor) within a single sample in two
minutes, which may have advantages over some other methods. This method is more representative of what the user inhales
as it is testing a heated liquid during exposure to the coil (potentially) catalytic surface, rather than only an unheated e-liquid.



## 5 Conclusion

In summary, the accelerated method described here is a suitable screening tool for rapid chemical assessment of heated e-liquids. It is a fairly recent (2014) recommendation by the EU TPD to assay heated e-liquids, and (to the best our knowledge)
there has only been one previously published study on the effects of aging/heating on e-liquids. We propose this method (with our recommended improvements) to be used as a standardized screening tool for e-liquids, and their aerosols, to identify potentially harmful chemicals, such as those recently banned in Australia or previously banned in Europe and the United Kingdom. With minor modification, this test could be used prior to importation or sale, to ensure that only tested products, containing approved ingredients, reach the consumer. In the absence of an approved therapeutic goods status for e-cigarettes,
the type of high-throughput testing described here is necessary as a minimal precaution to assess and reduce the potential harms of a consumer product that is generally accepted in the public to be a less harmful alternative to smoking.

CRediT statement: Natalie Anderson: data curation, formal analysis, investigation, methodology, project administration, visualization, writing – review and editing, writing – original draft.: Paul Pringle: data curation, formal analysis, investigation,
methodology, validation, writing – review and editing.: Ryan Mead-Hunter: formal analysis, investigation, writing – review and editing.: Benjamin Mullins: conceptualization, methodology, resources, writing – review and editing, funding acquisition.: Alexander Larcombe: conceptualization, methodology, project administration, resources, supervision, visualization, writing – review and editing, funding acquisition.: Sebastien Allard: data curation, formal analysis, investigation, methodology, resources, supervision, writing – review and editing, validation, visualization

Acknowledgements: This work was supported by funding from the Minderoo Foundation, Lung Foundation Australia, Cancer Council WA, and the Scottish Masonic Charitable Foundation

Declaration of Interests: All authors declare no competing interest

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




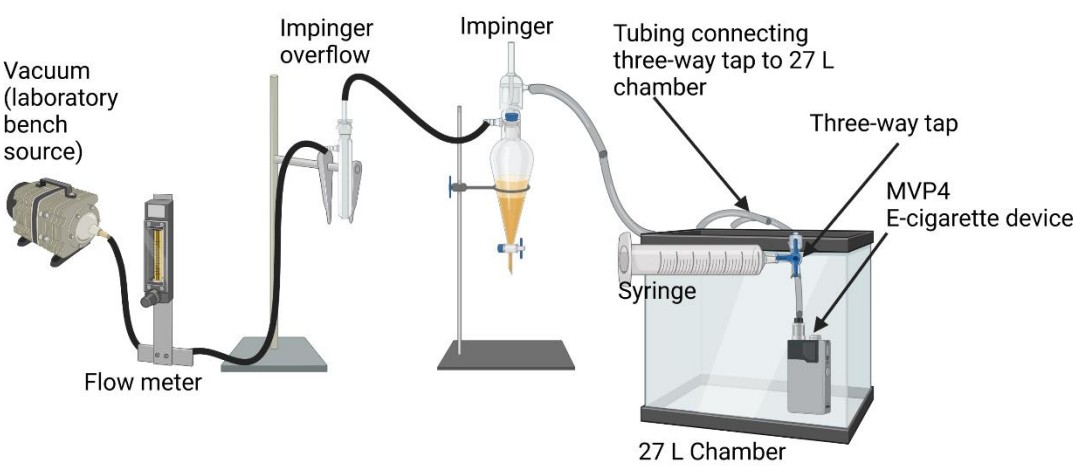

**Figure 1. Typical vaping set-up.** A vacuum drew air through the system at ~3 Lpm. The aerosol was drawn into a 60 mL syringe and a three-way tap turned to push the aerosol into the chamber. Air containing aerosol was drawn first into an impinger with 50:50 propylene glycol:glycerin base liquid.



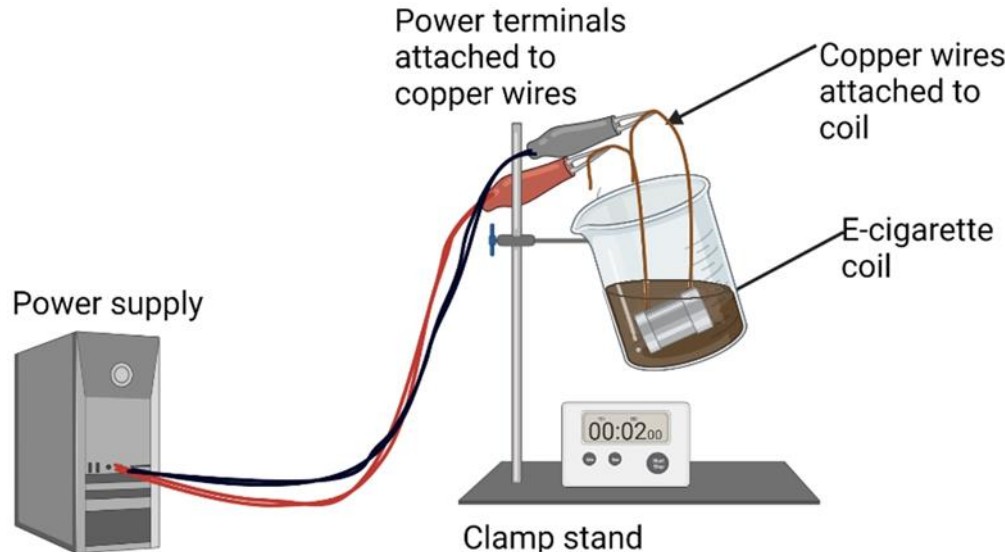


**Figure 2. Accelerated vaping set-up.** The power supply was attached to copper wires which were attached to an e-cigarette coil (Kanthal BVC) which was fully submerged at all times in e-liquid within a 100 mL beaker.



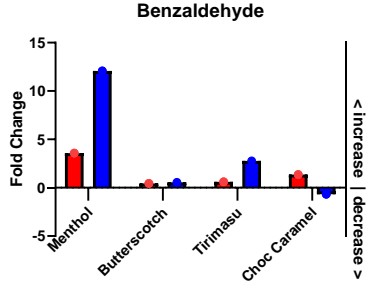

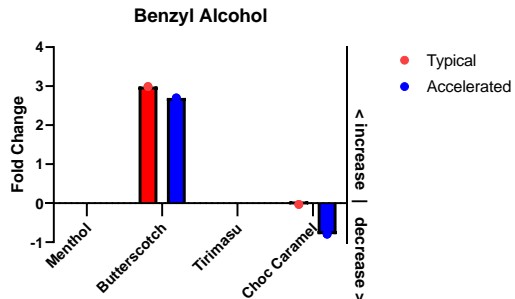

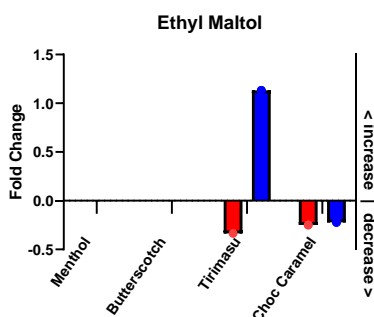

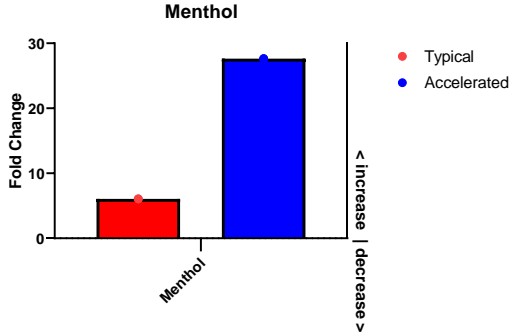

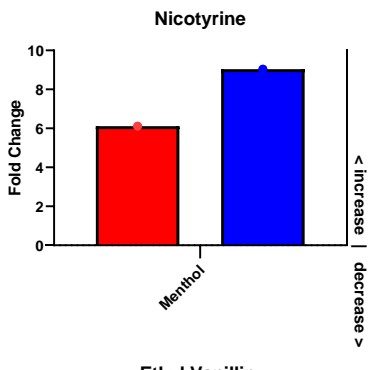

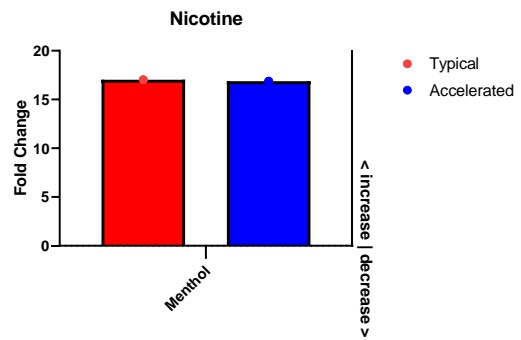

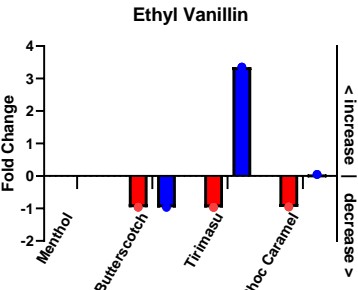

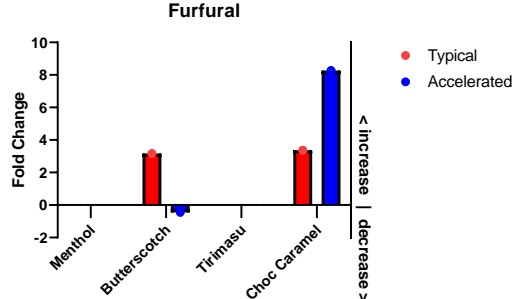



**Figure 3. Fold change comparison between heating methods.** E-liquid flavors are described on the X axis. Accelerated and Typical methods are indicated by the blue and red respectively. Y axis indicates the fold change compared to unheated i.e. Fold change = (Y-X)/X, where X is the unheated sample (mg/L concentration) and Y is the heated sample (mg/L concentration). Note different scales. Values that fall below the horizontal line at zero indicate a decrease in concentration from unheated sample, values above zero are an increase from unheated.


**Table 1. Assessment of 13 different chemical compounds from four e-liquids in both unheated and heated (accelerated and typical vaping methods).** Unheated sample is comprised of only e-liquid in "fresh" or un-vaped form. Heated sample for typical method assessed the leftover from the atomiser tank and accelerated method assessed heated e-liquid produced from a coil submerged in e-liquid. Chemicals are listed in alphabetical order. Blue shading = decrease from unheated sample, Orange

shading = increase from unheated sample. Acc = "accelerated" or "aged". u = undetected, italicized results indicate presence of an undetected sample and exclusion from analysis.

| Chemical Tested (mg/L) | Menthol unheated | Menthol Heated (acc) | Menthol Heated (typical) | Butter-scotch unheated | Butter-scotch Heated (acc) | Butter-scotch Heated (typical) | Tiramisu unheated | Tiramisu heated (acc) | Tiramisu heated (typical) | Choc Caramel unheated | Choc Caramel heated (acc) | Choc Caramel heated (typical) |
|---|---|---|---|---|---|---|---|---|---|---|---|---|
| Benzaldehyde | $9.87 \times 10^{-3}$ | $1.29 \times 10^{-1}$ | $4.51 \times 10^{-2}$ | 1.49 | 2.33 | 2.16 | $6.24 \times 10^{-1}$ | 2.48 | 1.06 | 1.49 | $2.81 \times 10^{-1}$ | 2.05 |
| Benzyl Alcohol | u | u | u | 13.40 | 49.53 | 53.46 | *345* | *u* | *285* | 238 | 49.53 | 231 |
| Ethyl Maltol | u | u | u | u | u | u | 339 | 723 | 225 | 891 | 693 | 669 |
| Ethyl Vanillin | u | u | u | 45.65 | 1.46 | 1.63 | 747 | 3250 | 23.73 | 1660 | 1740 | 76.98 |
| Eugenol | u | u | u | u | u | u | *$1.89 \times 10^{-1}$* | *u* | *$1.25 \times 10^{-1}$* | u | u | u |
| Furfural | u | u | u | $1.73 \times 10^{-1}$ | $9.38 \times 10^{-2}$ | $7.21 \times 10^{-1}$ | $6.57 \times 10^{-1}$ | *u* | $8.68 \times 10^{-1}$ | $5.16 \times 10^{-1}$ | 4.78 | 2.25 |
| Menthol | 32.54 | 932 | 230 | *u* | *30.18* | *11.83* | u | u | u | u | u | u |
| Nicotine | $4.77 \times 10^{-1}$ | 8.53 | 8.60 | u | u | u | u | u | u | u | u | u |
| Nicotyrine | $2.70 \times 10^{-4}$ | $2.71 \times 10^{-3}$ | $1.92 \times 10^{-3}$ | u | u | u | u | u | u | u | u | u |
| Thymol | u | u | u | u | u | u | u | u | u | u | u | u |
| Trans-cinnamaldehyde | u | u | u | u | u | u | u | u | u | *u* | *13.39* | *u* |
| 2-chloro-phenol | u | u | u | u | u | u | u | u | u | u | u | u |
| 4-(4-methoxyphenyl)-2-butanone | u | u | u | u | u | u | u | u | u | u | u | u |