# Peer review of "A rapid semi-quantitative screening method to assess chemicals present in heated e-liquids and e-cigarette aerosols"

_Aerosol Research, 2023_

## Author Comment (AC1)

Dear Reviewer 1,

Your comments and targeted feedback are greatly appreciated and we hope that the changes made improve the manuscript. Please find our responses below in bullet points.

**Authors should consider following changes:**

1. **Abstract: highlight introduction, methods, results so it is easier to follow**
   - This improvement has been made to the revised manuscript.
2. **Introduction:**
   - **the background of the scientific work is sound.**
     - Thank-you
   - **However, sentences are very long and sometimes confusing thus leading to difficulties in following paragraphs and/or understanding the motivation behind.**
     - The introduction has been considerably revised to reduce/eliminate lengthy sentences and substantially improve clarity
   - **Schematical representation of accelerated vs. typical would be highly beneficial.**
     - The authors have now amended the schematics to combine figure 1 and 2 as Figure 1 (a & b), and included a description of differences between methods which should allow/encourage better comparisons between methods. Mention of Figure 2 and 3 have been revised accordingly throughout.
   - **Line 65: why Kanthal was mentioned, and how it related to current work? It was not discussed later on.**
     - Kanthal (tradename for iron-chromium-aluminium alloy) was mentioned as the authors chose a kanthal BVC coil, and wanted to advise readers that degradation products can form even at low temperatures when exposed to a catalytic surface. However, as the reviewer observes, the discussion of this topic was unrelated to the manuscript goals, and therefore it is only been briefly mentioned in the revised manuscript – so that others can replicate our methodology if they choose: Line 71 now reads:

       "Importantly, chemicals present in e-liquids are known to degrade due to heating, either by boiling or evaporative-convection (depending on wetted-wick temperature) and this can be exacerbated by presence of catalytic surfaces such as Kanthal"

3. **Materials and methods:**
   - **what is the difference between e-liquid and e-cigarette? Terms are regularly interchanging, this can be very confusing**
     - The authors apologise for the confusion caused due to interchanging terms. The difference between terms has now been clarified in Line 30 of the manuscript to avoid confusion:

"Chemicals are present in the aerosol produced by the electronic cigarettes (e-cigarette) when the e-liquid it contains is heated and aerosolised. The e-cigarette is referring to the aerosol generating device, which uses the "e-liquid" to create aerosol by evaporation condensation method."

- **Figure 1: not clear what is meant by an atomizer**
  - Figure 1 (now Figure 1 a and b) has been updated (as below) to include label for e-liquid tank containing e-cigarette coil (the atomiser). It has also been clarified earlier on in the methods.

[Figure]

  -
- **Why flow of 3 L/min was selected?**
  - The flow volume was chosen for simplicity: it is just enough to keep a small flow moving through the system and standard flow meter devices in our lab measured up to 5 l/min.
- **Why the volume of 7.2L?**
  - The manuscript has now been updated on line 127 to state:

    "This process was repeated every minute for two hours (with the atomizer tank refilled after ~60 minutes), for 120 puffs total. While we acknowledge that vaping topography is extremely variable, 120 puffs over a 2 hour period (120 x 60 mL, puffs, therefore 7.2 L of inhaled air) was chosen to be representative of what a typical vaper might use".

- **Very nice catch to use 45° angle. The real lab pictures would be more clear than scheme**
- For clarity, we have chosen to use schematics to illustrate both the typical and accelerated aging equipment set-ups.

- **Why propylene glycol: glycerol mixture was selected and why in that ratio? Sources of materials were not mentioned**
    - The statement:

        "The propylene glycol: glycerol mixture was selected as 50:50 since it is a commonly sold ratio"

        has now been added to the methods section of the manuscript, line 148.

    - The materials were purchased from an online supplier. The authors preference was not to declare tradenames due to e-cigarette use in Australia being a contentious issue, and to avoid "naming and shaming".

4. **Results:**
    - **Table 1 is not uniform- decimal points sometimes 2, sometimes none, sometime logarithmic scale. If values are undetected (u), from where the reported values in italic are coming from? Not clear**
        - Thank-you, the authors apologise that this was not clear. The legend for Table 1 has been clarified regarding accuracy (which was to three significant digits) and amended accordingly (addition of underlined sentences) as follows:

        **"Table 1. Assessment of 13 different chemical compounds from four e-liquids in both unheated and heated (accelerated and typical vaping methods).** Unheated sample is comprised of only e-liquid in "fresh" or un-vaped form. Heated sample for typical method assessed the leftover from the atomiser tank and accelerated method assessed heated e-liquid produced from a coil submerged in e-liquid. Chemicals are listed in alphabetical order. Blue shading = decrease from unheated sample, Orange shading = increase from unheated sample. Acc = "accelerated" or "aged". U = undetected. Italicized results indicate that the compound was not present in one of the heating method samples and therefore fold change analysis was not possible for that chemical in that e-liquid flavor. Numbers are accurate to three significant figures (digits) and values less than zero are displayed with logarithmic scale for ease of reading."

        The text has also been adjusted (changes underlined) for clarity (from line 165);

        "Fold-change analyses were not possible on the following: one chemical was undetected in unheated form but detected in both heated forms (menthol in butterscotch tobacco); one chemical was undetected in unheated form and detected in only one heated form (trans-cinnamaldehyde in choc caramel). A further three chemicals were detected in unheated form and only one heated form (benzyl alcohol (Tiramisu), eugenol (Tiramisu) and furfural (Tiramisu)) (Table 1, represented by italicised values).

    - **Figure 3: e.g. on the graph for ethyl maltol, what is the value for menthol? non detected or very small so not visible? Should be stated.**
        - Thank-you for catching this. The graph has been amended so that if a chemical was not detected in a particular flavor (e.g. there was no ethyl maltol detected in the menthol or butterscotch flavor), then those flavors are not shown. The Figure legend has been changed to clarify this:

"If any chemical was not detected in a particular flavor then those flavors are not shown (e.g. there was no ethyl maltol detected in the menthol or butterscotch flavor)."

- **Figure 3 should be deleted from 3.2 title**
  - Thank-you. This change has now been made

5. **Discussion:**
   - **Sentences are too long and very hard to follow. Rewriting is needed**
     - Thank-you. The authors have revised the manuscript discussion for clarity and brevity.
   - **Authors should consider adding a picture of Rainout effect, it can nicely explain the observed events**

     - We do not have a high quality image that illustrates the rainout effect, however Figure 1 (a) and the figure legend has now been updated (as highlighted) to include the dimensions of the smaller tubing to allow easier visualisation. e.g. (changes underlined below):

       "**Figure 1. Vaping set-ups.** A. Typical vaping set up. A vacuum drew air through the system at ~3 Lpm. The aerosol was drawn into a 60 mL syringe and a three-way tap was turned to allow the syringe to push the aerosol through two ~4 mm ID and ~15 cm tubing lengths and into the 27 L chamber for mixing. Air containing aerosol was drawn first into an impinger with 50:50 propylene glycol:glycerin base liquid…"

       It has also been clarified in the method section line 123 as underlined:

       "To vape the device in line with recommendations by the Cooperation Centre for Scientific Research Relative to Tobacco (CORESTA), the ignition button was held for ~3 seconds, and the aerosol drawn from the device into a 60 mL syringe and then expelled through two ~ 4 mm ID, 15 cm tubing lengths into the custom made (27 L – 30x30x30 cm) chamber, using a three-way tap (Dispoflex™, Disposafe health and life care Ltd, Haryana, India) (Cooperation Center for Scientific Research Relative to Tobacco (CORESTA) (2015))."

   - **What is the solubility of mentioned compounds in impinger fluid? The effect of solubility is mentioned however not thoroughly discussed, even though it is very relevant**
     - It would be difficult to obtain solubility for each compound we are assessing, and the study was not designed to investigate this. As such the discussion paragraph beginning on line 220 has been reworded to acknowledge this as a limitation and consideration for future studies (changes underlined):

       "The remaining two discrepancies involved furfural (1/11) and benzaldehyde (1/11), and these compounds were found in increased quantities with the "typical" method compared to the "accelerated" method. Because these two discrepancies contain low molecular weight/low boiling point products we suspect they may have evaporated more readily

(compared to the "typical" method) and that our "accelerated" method was simply unable to capture compounds with low boiling point that volatise easily (Erythropel et al., 2019). For example, furfural has the lowest boiling point and molecular weight of all chemicals detected (162°C, 96.09 g/mol) and was detected as decreased from the unheated sample in both "butterscotch" and "tiramisu" flavor. While our study design angled the beaker at 45° to allow re-condensation of any vapor on the beaker wall, the experiment was carried out in a ventilated fume hood, which may have increase the loss of highly volatile compounds. In future study, the addition of a lid on the angled beaker as well as monitoring the liquid temperature on the wick (or surrounding the wick), may help to reduce discrepancies to allow full validation of the method for detection of aerosols (Li et al., 2021, Palazzolo, Caudill, Baron, & Cooper, 2021, Bitzer et al., 2018). Further studies should also be designed to consider the solubility of the flavoring compound in the base excipients (i.e. propylene glycol or glycerol), and not the boiling point, as this is suggested to be a major indicator of whether a compound will be detected in/carried over into an aerosol (Erythropel et al., 2019)."

- **What is wetted wick T and why it is relevant?**
   - The authors apologise for the confusion regarding the term wetted-wick, the wetted wick temperature was not directly relevant, but rather the liquid temperature on the wick. Terminology has now been clarified. The authors thought measuring the liquid temperature on the wick may be helpful to elucidate whether aerosol formation was by boiling or evaporative convection, and therefore production of low boiling point chemicals likely or not. The discussion has been clarified in the revised version.

---

## Author Comment (AC2)

Dear Reviewer Two,

Thank-you for your review of our manuscript and targeted comments. We believe the changes we have made based on your perspective and comments improve the manuscript.

**The manuscript describes a novel and accelerated method to determine the chemical composition of liquids for e-cigarettes before and after heating. The method uses an e-cigarette coil, submerged in the e-liquid to be analyzed. The liquid is filled into a small beaker, which is kept tilted at an angle of 45° so that aerosolized droplets would be collected on the beaker wall and run back into the liquid reservoir. Eventually, the remaining liquid in the reservoir is chemically analyzed by GC-MS. I have to admit that for me as an aerosol scientist, who is not very familiar with the characterization of e-liquids and e-cigarettes, the manuscript is difficult to follow.**

- • Thank-you for this feedback. We have considerably revised the manuscript for clarity and brevity and hope that this greatly improves the ability to follow the manuscript.

**I am particularly struggling with understanding, why (according to the title) the paper is supposed to describe a quantitative method to analyze e-cigarette** aerosols**.**

Thank-you for addressing this concern and for the opportunity to improve the manuscript by clarifying the aerosol detail as it was not conveyed clearly. The following description has been added now on line 104 of the manuscript to describe our premise more clearly behind the "accelerated method" and how it might represent the aerosol:

"Our premise for collection of e-cigarette aerosols in the liquid was as follows:

1. An e-cigarette is an evaporation condensation aerosol generator – intended to modify the e-liquid as little as possible during aerosolization, however, it does thermo-oxidise, hence the need for this research;
2. Our "accelerated" method of a heating the e-liquid via a submerged coil creates a "bubbling aerosol generator" (Vidamantas, 1997). Like an evaporation aerosol condensation generator, a bubbling generator will modify the e-liquid minimally, however, may allow more volatile compounds to preferentially aerosolise;
3. The creation of an aerosol via bubbling can allow aerosol capture either whilst bubbling through the bulk liquid (when cooling) or at the gas-liquid surface (Ghiaassiaan, 1997 and Koch, 2012);
4. Surface bubbles can generate aerosol either by jet or film droplets when they burst, and based on combinations of surface tension and bubble size, aerosol will recombine with the liquid the bubble arises from when it bursts (Koch, 2012, Mead-Hunter, 2018).
5. Thereby, a combination of these processes should ensure we retain a representative sample of the same material that is aerosolised, as well as possibly more of the thermos-oxidised (aged) material we are interested in."

**The premise of the study was to produce an e-cigarette aerosol that would be immediately captured. Eventually, only liquids are analyzed and I miss a description why the analyzed liquids are supposed to be representative for the emitted aerosol (I assume that this is the goal). The authors should focus the description of the methodology more towards the aerosol characterization, bearing in mind that the journal is on "Aerosol Research".**

- Thank-you for this observation. We hope that the above description and the related modification to the manuscript on line 104 greatly improves the description of how the analysed liquids are representative of the aerosol.

**I also wonder how "quantitative" the method really is, considering that the fold-change comparison in Figure 3 reveals rather large discrepancies of the two methods and nothing is mentioned about the accuracy and recovery rate of the reference method. Shouldn't the method rather be termed "qualitative" or semi-quantitative"? For a quantitative method, I would expect to see more analyses on the method's uncertainty.**

- Thank-you for this observation regarding the accuracy of the title wording "quantitative". The title has been changed to semi-quantitative.

**Specific comments:**

- **Line 91: "vacuum" should read "vacuum pump" (also applies to figure 1)**
    - The image of a vacuum pump was a stand-in for the lab bench vacuum source. Figure one has now been changed to avoid confusion. Please see below:

[Figure]

- **Line 91/92: model number of the flow meter is missing**
    - our apologies for missing this detail it has now been added on line 90.
- **Line 99: If the flow rate is 3 l/min and the total duration 4 h (240 minutes) then the total volume should be 720 l, not 7.2 l**
    - Apologies it was not clear was the 7.2 L was referring to. This has been clarified in the revised manuscript (line 97) which shows that we refer to 7.2L of e-cigarette aerosol, not 7.2L of total air flowing through the system;

    "This process was repeated every minute for two hours (with the atomizer tank refilled after ~60 minutes), for 120 puffs total.  While we acknowledge that vaping topography is

extremely variable, 120 puffs over a 2 hour period (120 x 60 mL, puffs, therefore 7.2 L of inhaled air) was chosen to be representative of what a typical vaper might use (Etter, 2014).

- **Line 110/111: What is meant with "aerosol (…) would recondense on the wall". As far as I understood, the goal is to collect only droplets and no (condensed) vapors. Do you mean that aerosolized droplet would collide with the wall and flow back into the reservoir? On the other hand, how certain is it that vapors do not recondense? And what is the efficiency for the collection of the droplets on the wall? I assume that this would be only based on impaction. Any information on the droplet size, velocity and Stokes numbers?**
    - The authors were not able to measure this mode of collection with certainty and calculation of capture rates by impingement on the beaker wall would include many assumptions. This is recognised as a limitation of the study on line 225 to 230.
- **Line 280/281: Here, the authors claim that the method potentially (!) allows for analyzing resultant aerosols.**
    - Theoretically we were able to capture resultant aerosols as is now clarified on line 104 however some limitations of the method were identified insection 4.1, therefore the method only "potentially" captured aerosols, pending minor modifications to the described method. We have therefore changed the word potentially on line 284 to the phrase "with minor modifications", to more accurately represent what the authors were intending to convey.
- **Figure 3: There is a typo in the caption of most axes: "Tirimasu" should read "Tiramisu"**
    - Thank-you this has now been corrected.